

# Bite force data suggests relationship between acrodont tooth implantation and strong bite force

Kelsey M. Jenkins and Jack O. Shaw

Department of Earth and Planetary Sciences, Yale University, New Haven, United States of America

## ABSTRACT

Extant and extinct reptiles exhibit numerous combinations of tooth implantation and attachment. Tooth implantation ranges from those possessing roots and lying within a socket (thecodonty), to teeth lying against the lingual wall of the jawbone (pleurodonty), to teeth without roots or sockets that are attached to the apex of the marginal jawbones (acrodonty). Attachment may be ligamentous (gomphosis) or via fusion (ankylosis). Generally speaking, adaptative reasonings are proposed as an underlying driver for evolutionary changes in some forms of tooth implantation and attachment. However, a substantiated adaptive hypothesis is lacking for the state of acrodont ankylosis that is seen in several lineages of Lepidosauria, a clade that is plesiomorphically pleurodont. The convergent evolution of acrodont ankylosis in several clades of lepidosaurs suggests a selective pressure shaped the evolution of the trait. We hypothesize that acrodont ankylosis as seen in Acrodonta and *Sphenodon punctatus*, is an adaptation either resulting from or allowing for a stronger bite force. We analyzed bite force data gathered from the literature to show that those taxa possessing acrodont dentition possess a stronger bite force on average than those taxa with pleurodont dentition. Dietary specialists with pleurodont dentition may also possess relatively high bite forces, though body size may also play a role in their ability to bite hard. Furthermore, our results have implications for the evolution of acrodont ankylosis and potential behaviors related to strong bite force that influenced the evolution of acrodonty within Acrodonta and Rhynchocephalia.

# INTRODUCTION

Acrodont tooth implantation, where the tooth sits at the summit of the tooth-bearing bone, evolved multiple times within Lepidosauria. It appears at least twice within squamate reptiles, as seen in Acrodonta (*Romer, 1956*) and Trogonophidae (*Gans, 1960*), and once within Rhynchocephalia (*Jenkins et al., 2017*) (Fig. 1). In Acrodonta and *Sphenodon punctatus,* the only living representative of Rhynchocephalia, the dentition is strongly ankylosed (i.e., fused) via the adjacent bone. In those taxa, teeth and surrounding tissues have been investigated thoroughly via histological studies (*Cooper & Poole, 1973*; *Smirina & Ananjeva, 2007*; *Kieser et al., 2009*; *Kieser et al., 2011*; *Haridy, 2018*), CT data (*Dosedělová*

Corresponding author
Kelsey M. Jenkins,
kelsey.jenkins@yale.edu

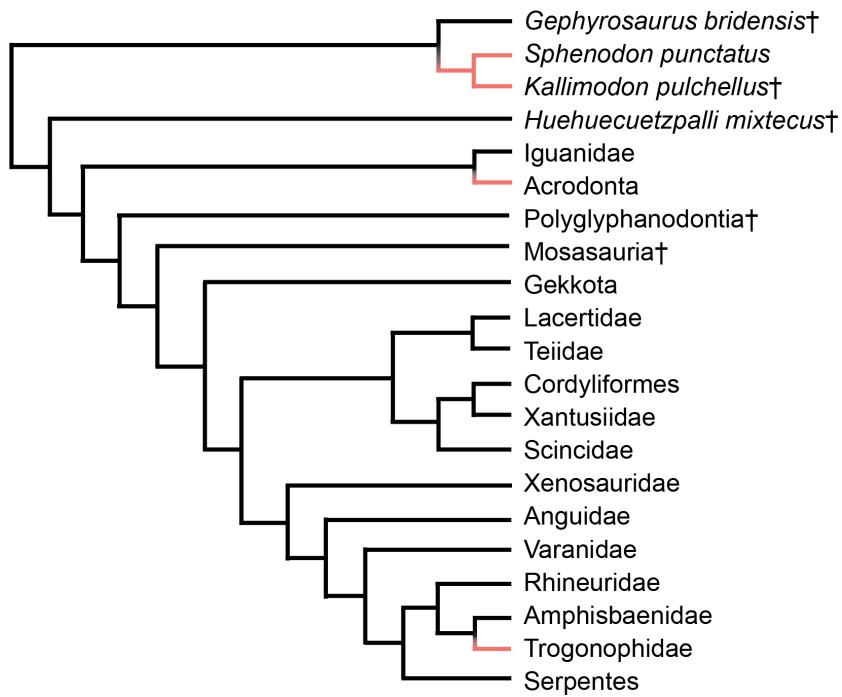

**Figure 1** **Simplified phylogeny of Lepidosauria from** *Gauthier et al. (2012)*. Orange branches indicate acrodont ankylosis. Phylogeny is based on a morphological dataset and parsimony analysis.

*et al., 2016*), and in vitro staining (*Buchtová et al., 2013*; *Salomies et al., 2019*). However, the evolution of acrodont tooth implantation is seldom discussed in an adaptive context.

*Smith (1958)* suggested that acrodonty is a trait associated with anchoring a permanent dentition, and possibly inhibiting tooth replacement. Presently, this remains the only functional hypothesis associated with acrodont dentition. Furthermore, growing body of work shows that anchoring dentition does not inhibit tooth replacement, and anchoring occurs after tooth generation and replacement cease (*Handrigan & Richman, 2010*; *Buchtová et al., 2013*; *Dosedělová et al., 2016*; *Haridy, 2018*; *Salomies et al., 2019*). *Smith (1958)* also suggested that thecodonty, where the tooth sits within a socket, is associated with permanent dentition. While that is the case in mammals, it is well known that several reptilian lineages and early synapsids possess thecodont implantation and replace their teeth with some regularity (e.g., *Edmund, 1960*; *Edmund, 1962*; *Gaengler, 2000*; *McIntosh et al., 2002*; *LeBlanc et al., 2017*; *D'Emic et al., 2019*; *Snyder et al., 2020*). Unlike acrodont dentition, thecodonty is not necessarily associated with ankylosis and may attach to the surrounding bone ligamentously, termed gomphosis (*Osborn, 1984*).

There are a suite of traits commonly associated with acrodont tooth implantation, most typically reduced tooth counts and severe tooth wear (*Augé, 1997*; *Haridy, 2018*). However, those characters are not necessarily associated with every taxon exhibiting acrodont tooth implantation. Though some have noted a loss of the alveolar foramen in the teeth of acrodontan squamates, using this trait to diagnose those taxa (*Zaher & Rieppel, 1999*), it was later found that *Pogona vitticeps* possesses nutrient foramina supplying the pulp

cavities (*Haridy, 2018*). A slowing or lack of tooth replacement, called monophyodonty, is also commonly associated with acrodont tooth implantation (*Smith, 1958*; *Cooper, Poole & Lawson, 1970*), although exceptions do exist (*Gow, 1977*; *Haridy, LeBlanc & Reisz, 2018*). Even with monophyodont dentition, additional teeth are typically still added to the posterior end of the tooth row throughout ontogeny, as is the plesiomorphic condition within Reptilia (*Robinson, 1976*; *Gow, 1977*; *Rieppel, 1992*; *Reynoso, 2003*).

As individuals of Acrodonta and *Sphenodon punctatus* age, the boundary between tooth and bone becomes difficult to determine externally (Fig. 2). This is a result of alveolar bone growing to surround the outer portion of the tooth through ontogeny (*Buchtová et al., 2013*; *Haridy, 2018*). This feature caused some to erroneously propose that *S. punctatus* lacks teeth entirely, instead possessing a serrated jawbone (*Mlot, 1997*). Severe wear may obscure the anterior dentition in older, acrodont, monophyodont lepidosaurs, and in some cases the teeth may be worn to the point where the bone itself forms the occlusal surface in the anterior portion of the mouth (*Robinson, 1976*). To resist wear as the reptile ages, the pulp cavity infills with bone and secondary dentine as seen in members of Acrodonta (*Throckmorton, 1979*; *Smirina & Ananjeva, 2007*; *Dosedělová et al., 2016*; *Haridy, 2018*) or secondary dentine and pulp-stones as seen in *S. punctatus* (*Kieser et al., 2009*).

The ancestral state of tooth implantation and attachment in the reptilian lineage is thought to involve a tooth set in a shallow socket (i.e., subthecodonty) attached via ankylosis (*Bertin et al., 2018*), though some of the most basal reptiles exhibit pleurodont tooth implantation (*LeBlanc & Reisz, 2015*). Furthermore, the periodontal ligament is likely ancestrally present in all amniotes (*LeBlanc et al., 2016*). However, reptiles have since explored many forms of tooth implantation (acrodonty, pleurodonty, and thecodonty) and attachment (ankylosis and gomphosis) in varying combinations. Adaptive interpretations are occasionally used to explain why reptiles may stray from the ancestral state within their respective clades (*Smith, 1958*; *Noble, 1969*; *Osborn, 1984*). Other adaptations for attachment include dentine infoldings, called plicidentine, which evolved independently multiple times within Reptilia, and it is interpreted to be a mechanism to strengthen tooth attachment in kinetic-feeding predators by increasing the surface area for attachment (*Preuschoft et al., 1991*; *Maxwell, Caldwell & Lamoureux, 2011*; *Brink, Leblanc & Reisz, 2014*; *MacDougall, LeBlanc & Reisz, 2014*). Even the loss of teeth may be associated with the evolution of other adaptive structures, like a keratinous beak (*Davit-Béal, Tucker & Sire, 2009*).

The ancestral state of tooth implantation and attachment for crown lepidosaurs is likely pleurodont ankylosis, seen in basal members of both Squamata and Rhynchocephalia (e.g., *Evans, 1980*; *Whiteside, 1986*; *Reynoso, 1998*; *Simões et al., 2018*). Additionally, the lepidosauromorphs *Marmoretta* and *Sophineta* possess pleurodont tooth implantation (*Evans, 1991*; *Evans & Borsuk-Białynicka, 2009*). The evolution of acrodont ankylosis accompanied by bone and secondary dentine deposition, as seen in Acrodonta and *Sphenodon punctatus,* lacks any adaptive hypothesis. Here we suggest that this combination of traits is an adaptation associated with strong bite force. Anecdotal evidence suggests that acrodont taxa possess a strong bite: *S. punctatus* is said to possess a painful and 'vice-like' bite (*Robb, 1977*; *Daugherty & Cree, 1990*), and one of the authors (KMJ) notes from

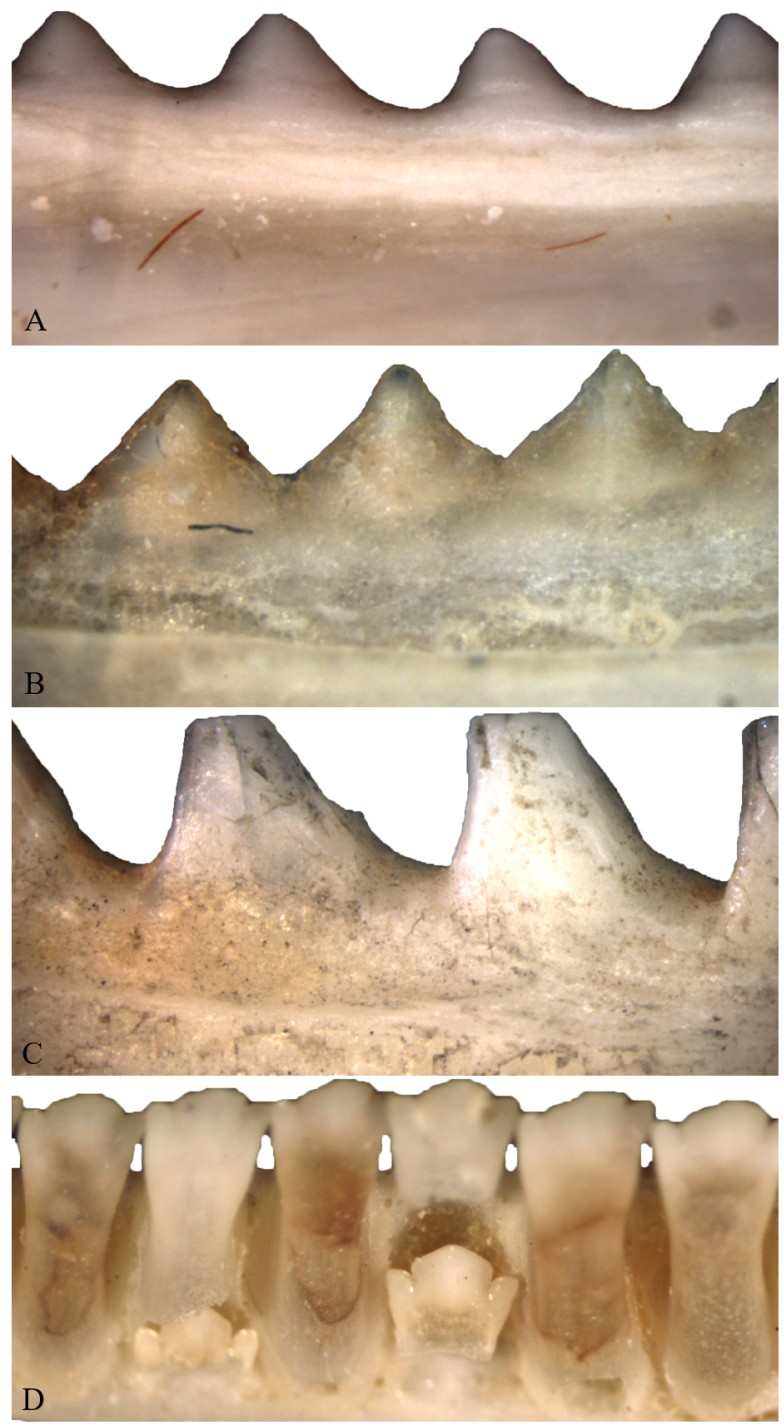

**Figure 2 Acrodont ankylosis as seen in two species of Acrodonta (A & B) and a rhynchocephalian (C) and pleurodont ankylosis (D).** (A) the chameleon *Fercifer oustaleti* YPM R 1214, (B) the agamid *Agama agama* YPM R 17936, and (C) the rhynchocephalian *Sphenodon punctatus* YPM R 10647. (D) Pleurodont tooth implantation as seen in *Ctenosaura* sp. YPM R 11060.

personal experience that the veiled chameleon, *Chamaeleo calyptratus*, also possesses a painful bite and is reluctant to release its victim. Bite-force analyses also indicate agamids have a stronger bite than *S. punctatus*, relative to body size (*Schaerlaeken et al., 2008*; *Jones & Lappin, 2009*).

The literature concerning bite force in lepidosaurs is numerous and implies a multitude of benefits for increased bite force. For instance, increase bite force is thought to improve prey capture and handling in lepidosaurs, particularly for the consumption of hard-bodied prey (*Herrel et al., 1999*; *Herrel et al., 2001*; *Verwaijen, Van Damme & Herrel, 2002*; *Meyers, Nishikawa & Herrel, 2018*). High bite force may also aid in territory defense and dominance (*Herrel, Meyers & Vanhooydonck, 2001*; *Lailvaux et al., 2004*; *Huyghe et al., 2005*; *Husak et al., 2006*; *Jones & Lappin, 2009*), and mating success (*Lappin & Husak, 2005*; *Husak, Lappin & Van Den Bussche, 2009*; *Herrel et al., 2010a*; *Herrel et al., 2010b*). Higher bite force in lizards is often accompanied by skeletal correlates in the cranium and increased mass of the adductor musculature compared to those with lower bite force (*Herrel, McBrayer & Larson, 2007*; *Fabre et al., 2014*). Cranial kinesis also plays a strong role in bite force, with a more rigid or akinetic skull being more capable of producing a strong bite (*Erickson, Lappin & Vliet, 2003*; *Wroe, McHenry & Thomason, 2005*; *Tseng & Binder, 2010*; *Cost et al., 2020*). Thus, the varying degrees of kinesis in lizard and tuatara skulls can certainly impact bite force within Lepidosauria (*Frazzetta, 1962*).

We suggest that acrodont tooth implantation is yet another skeletal trait associated with bite force. We hypothesize that taxa possessing acrodont dentition also possess a higher bite force, compared to those with pleurodont dentition, relative to body size. Furthermore, the accompanied bone deposition around the base of the dentition may also assist in resisting strong biting. In order to test our hypothesis, we analyzed bite force data based on a comprehensive literature review among lepidosaurian taxa. We found that size-normalized bite force was significantly greater in acrodont lepidosaurs than pleurodont lepidosaurs. Furthermore, we discuss the evolution of acrodont ankylosis within an adaptive context in response to high bite force.

## MATERIALS & METHODS

To assess the relationship between lepidosaurian bite force and tooth implantation, we analyzed previously recorded bite force data. We collected mean snout-vent length (SVL), mean head depth (HD), and mean bite force (BF) measurements from thirty-nine peer-reviewed papers (Supplementary Files). Though bite force can be measured via different methods (see *Lappin & Jones, 2014*), all studies analyzed here measured the orthal bite. Following previous studies (*Erickson et al., 2004*; *Wroe, McHenry & Thomason, 2005*; *Sellers et al., 2017*) we analyzed log-transformed measurements (analyses of non-transformed data provided in supplement).

We focused on the relationship between SVL and BF, as SVL is the most commonly reported measure of size in reptiles (Fig. 3). However, many squamate reptiles possess elongate body forms that are not necessarily correlated to cranial allometry, and thus may not strongly correlate with bite force. Because of this, we also standardized by head

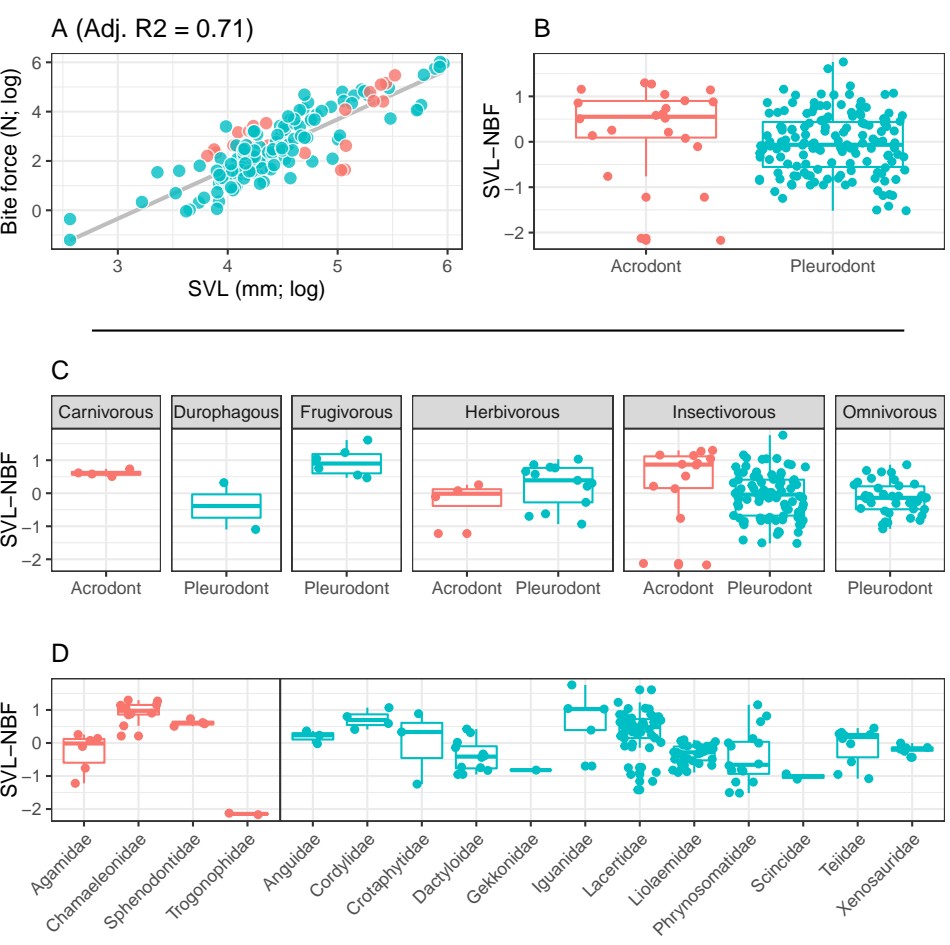

**Figure 3 Acrodont vs. Pleurodont bite force (SVL).** (A) Linear regression (grey line) of log-transformed snout-vent length (SVL) and bite force. (B) Boxplot of the distributions of snout-vent length normalized bite force (SVL-NBF), calculated as the residual values from the linear regression in (A), overlain with residual values as datapoints. (C) Breakdown of SVL-NBF values show in (B) by diet. (D) Breakdown of SVL-NBF values shown in (B) by family and separated based on tooth implantation (acrodonts in red, pleurodonts in blue).

depth in separate analyses (Fig. 4). Multiple studies evaluating lepidosaurian bite force suggest that head depth is a good predictor of bite force because it accommodates the adductor musculature (*Herrel, De Grauw & Lemos-Espinal, 2001*; *Lappin, Hamilton & Sullivan, 2006*; *McBrayer & Anderson, 2007*; *Herrel et al., 2010a*; *Herrel et al., 2010b*). Tooth implantation was assessed by the authors.

To examine differences in bite force between acrodont and pleurodont taxa, analyses of covariance (ANCOVA) were performed using both size variables (log-SVL and log-HD) as covariates. To further compare bite force across taxa of significantly different body masses, we calculated normalized bite force (NBF) as the residuals of a linear regression fit to (1) log-SVL and log-BF or (2) log-HD and log-BF. We refer to these values as SVL-NBF and HD-NBF, respectively. Differences in NBF between tooth implantation groups were then assessed using Kolmogorov–Smirnov (KS) tests.

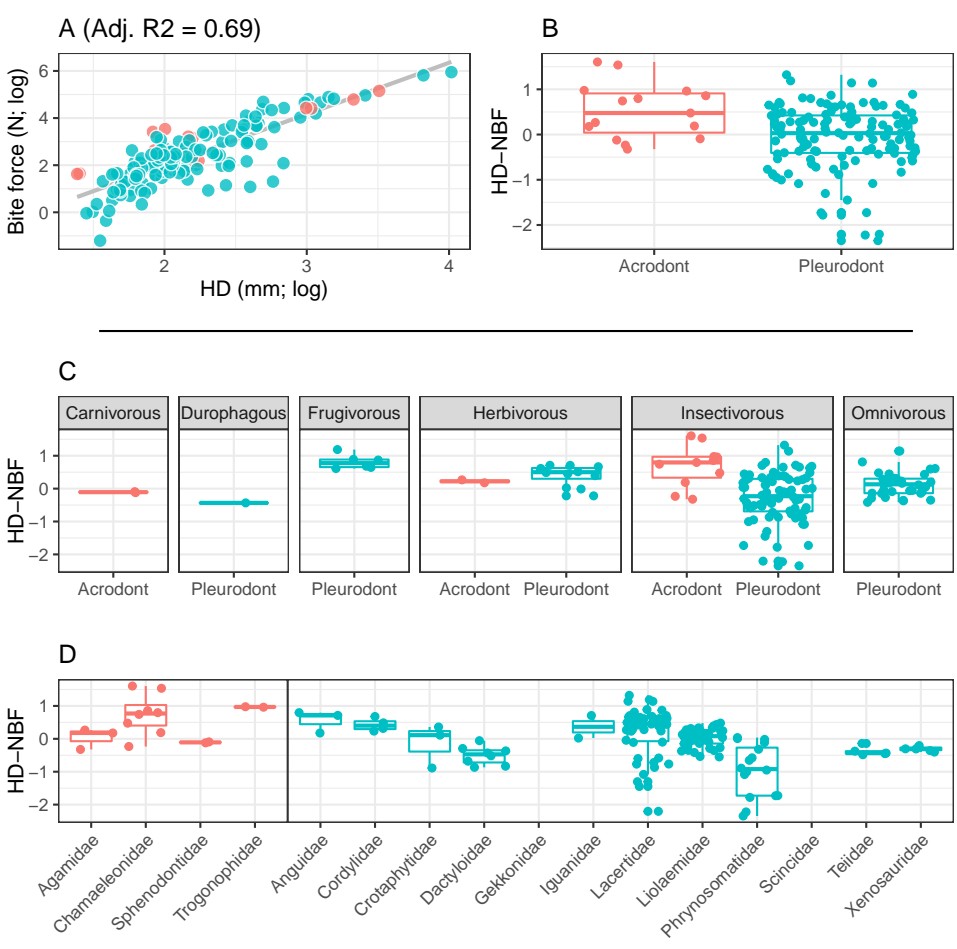

**Figure 4  Acrodont vs. Pleurodont bite force (HD).** (A) Linear regression (grey line) of log-transformed head depth (HD) and bite force. (B) Boxplot of the distributions of head depth normalized bite force (HD-NBF), calculated as the residual values from the linear regression in (A), overlain with residual values as datapoints. (C) Breakdown of HD-NBF values show in (B) by diet. (D) Breakdown of HD-NBF values shown in (B) by family and separated based on tooth implantation (acrodonts in red, pleurodonts in blue).

To evaluate the proportion of the lepidosaurian tree that has been examined in terms of bite force, we tallied all known publications that record lepidosaurian bite force (Fig. 5; Supplemental Information). This includes those publications that were not included in the initial analyses that compare bite force between acrodont and pleurodont taxa due to a lack of raw bite force data or a lack of necessary variables (i.e., SVL). Seventeen lepidosaurian families were represented by bite force data, including four acrodont families and 13 pleurodont families. Dactyloidae was represented by the most species ($n = 49$), while the families of Sphenodontidae, Varanidae, and Trogonophidae were only represented by single species.

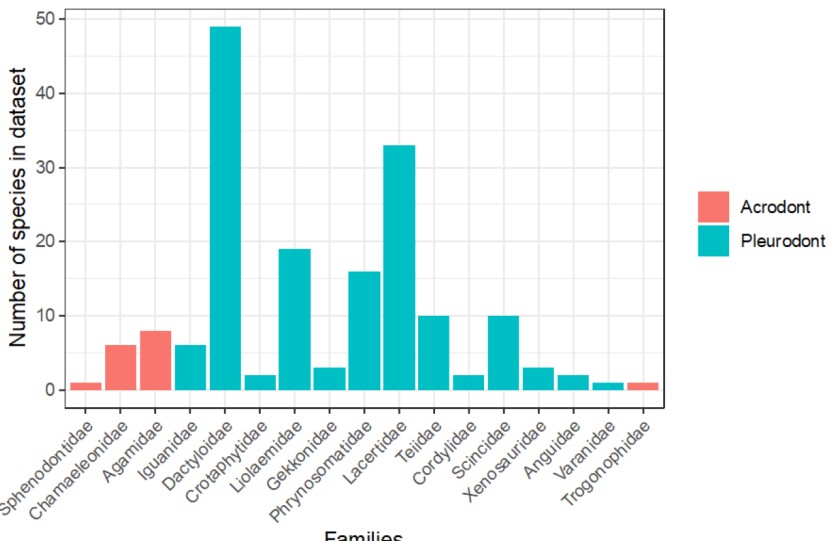

**Figure 5** **Number of species analyzed for bite force by family, colored by tooth implantation.**

## RESULTS

Bite force is higher in acrodont taxa than in pleurodont taxa after accounting for size differences (Figs. 3B & 4B). Raw bite force values ranged from 0.3 to 409.3 Newtons, SVL ranged 13.0–389.0 mm, and HD ranged 4.0–55.5 mm. SVL-NBF ranged −2.17 to 1.76, whereas HD-NBF ranged −2.35 to 1.61 (Supplemental Information). ANCOVAs of tooth implantation type and SVL and of tooth implantation type and HD have low $p$-values (0.064 and 0.0023, respectively) indicating differences in bite force between the acrodont and pleurodont taxa after accounting for SVL and HD. According to one-sided KS tests, acrodont SVL-NBF and HD-NBF values were significantly greater than those of pleurodonts. Linear regressions of log-SVL and log-BF, and of log-HD and log-BF were statistically significant and exhibited positive slopes ($p$-value <0.05). Correlations were stronger between log-SVL and log-BF (Adj R-square = 0.71), compared to log-HD and log-BF (Adj R-square = 0.69).

The only direct comparisons that could be made for both tooth implantation categories and diet were for insectivory and herbivory (Figs. 3C & 4C). According to one-sided KS tests, SVL-NBF and HD-NBF values were significantly greater for insectivorous acrodonts compared to insectivorous pleurodonts ($p$-value < 0.05). Overall, pleurodont insectivores exhibited a large range of NBF values (Fig. 3C). Although acrodont insectivores seemingly also exhibited a wide range of SVL-NBF values, this is influenced by the elongate body plan seen in on taxon, *Trogonophis wiegmanni*, in which head dimensions do not correlate strongly with SVL (Fig. 3C). HD-NBF values for acrodont insectivores range less than the SVL-NBF values of the same group (Fig. 4C). A direct comparison of herbivorous acrodonts and pleurodonts reveals a lack of significant difference between the NBF vales of the two groups, according to one-sided KS tests. While direct comparisons between tooth implantation types and other diets are not possible using the available data, we also found

that pleurodont frugivores exhibit the highest median NBF values whereas durophagous pleurodonts exhibit the lowest median NBF values, although the latter is based on a small number of measurements ($n = 2$).

Low SVL-NBF values in Trogonophidae indicate that the clade exhibits lower bite force than expected for SVL (Fig. 3D). These values were much lower than for other acrodont taxa, dramatically impacting the range and median SVL-NBF values for acrodonts. This trend is not present in HD-NBF, in which Trogonophidae exhibits the highest median bite force (Fig. 4D). Excluding Trogonophidae, Chamaeleonidae exhibited the highest SVL-NBF and HD-NBF values among acrodonts. Among pleurodont taxa, Lacertidae exhibits the largest range of NBF values for both SVL and HD. Anguidae exhibits the greatest median values for HD-NBF. Iguanidae exhibits the greatest median SVL-NBF values. Both Gekkonidae and Scincidae exhibit low median SVL-NBF values, but data does not exist for either clade for HD-NBF. Phrynosomatidae exhibits the lowest median HD-NBF values.

## DISCUSSION

Thus far, anatomical research related to bite force in lepidosaurian reptiles has focused primarily on cranial musculature and skeletal dimensions, namely head depth, length, and width. However, teeth are more intimately associated with biting and oral processing than the aforementioned elements. Dental morphology is often adapted for diet, with generalists possessing a more unspecialized dentition and specialists often possessing more unique tooth morphologies (e.g., *Estes & Williams, 1984*), though true specialists within lizards are rare and diets are often quite varied (*Greene, 1982*; *Schaerlaeken et al., 2012*). It should come as no surprise that tooth implantation and attachment are also shaped by oral processing capabilities. For example, multiple functional hypotheses exist for the evolution of thecodonty and associated periodontal ligament: a means of shock absorption and dissipation (*Noble, 1969*; *Picton, 1989*; *McIntosh et al., 2002*; *Bosshardt et al., 2008*), facilitation of post-eruption tooth movement (*Osborn, 1984*; *Bosshardt et al., 2008*), creation of a sensory system to allow the jaws to reposition during mastication (*Bosshardt et al., 2008*), flexible attachment of tooth to bone (*LeBlanc & Reisz, 2013*), and for anchoring permanent dentition in mammals as mentioned above (*Smith, 1958*). Similar hypotheses for the evolution of acrodont ankyloses in reptiles are lacking. Here we show that there is a relationship between acrodont ankylosis and high bite force. However, there is still the question of whether (1) acrodont ankylosis developed due to strong bite force, or if (2) strong bite force evolved as a consequence of acrodont ankylosis.

In the first scenario, acrodont ankylosis is a response to increased bite force by further securing the tooth to the bone as a means to resist failure during strong biting. Previous work shows that a stouter, blunter tooth, like that of acrodont taxa, is more resistant to failure under increased bite forces (*Lucas & Luke, 1984*; *Evans & Sanson, 1998*; *Jones, 2006*), compared to a more columnar or piercing tooth seen in most pleurodont lepidosaurs which is prone to breakage under increased forces (*Evans & Sanson, 1998*; *Erickson, Lappin & Vliet, 2003*). We suggest that the ankylosis and bone deposition seen in Acrodonta and

*Sphenodon punctatus* that accompanies the typical acrodont tooth morphology would also aid in resisting tooth failure. Simply put, a fused tooth is sturdier than a tooth attached via soft tissue. The specific combination of morphology, implantation, and attachment seen in Acrodonta and *S. punctatus* allows for a tooth that is most resistant to failure. However, that is not to say that breakage is impossible in taxa possessing acrodont ankylosis. The extremely strong adherence of teeth can result in the occasional breakage of both tooth and bone (*Dosedělová et al., 2016*). Perhaps this is due to the more brittle nature of such a strongly fused tooth, compared to a ligamentous attachment in which the tooth is better cushioned during feeding (*Noble, 1969*; *Picton, 1989*; *McIntosh et al., 2002*; *Bosshardt et al., 2008*). Though pleurodont dentition in other lizards is also ankylosed, most lack the bone growth that adheres the tooth to the jaw that is seen in Acrodonta and *S. punctatus*. However, there are a few notable squamates without acrodont dentition that possess large deposits of boney tissue around the base of the teeth (e.g., *Caldwell, 1999*; *Zaher & Rieppel, 1999*).

In the second scenario, strong bite force is a response to acrodont ankylosis. Acrodonta and *Sphenodon punctatus* are monophyodont and exhibit severe wear, particularly in the anterior dentition as seen in older individuals. Although those individuals have extremely worn teeth, they still manage to capture and consume prey. If the dentition is severely worn due to a lack of replacement, increased bite force would be crucial in allowing the jaws to clamp tightly onto prey. However, this could be a factor of monophyodonty instead of acrodonty alone. Nonetheless, older individuals with few functional teeth can still forage and consume as needed. If strong bite force in Acrodonta and *S. punctatus* evolved as a mechanism to aid in territory defense or increased mating success (opposed to prey handling), an older animal may be successful even though it possesses severely worn teeth. At this time, we cannot favor one hypothesis over another. It is also possible that different lineages acquired acrodont ankylosis under either scenario.

Two other hypotheses unrelated to increased bite force could explain the evolution of acrodont ankylosis from an initially pleurodont state. The first is that this combination of implantation and attachment evolved convergently in response to a shared diet. All extant lepidosaurs possessing acrodont ankylosis fill various dietary niches ranging from insectivory to herbivory (Figs. 3 and 4), calling to question the idea that the combined traits are currently acting as an adaptation for similar diets. Furthermore, extant squamates eating hard-shelled organisms, such as *Varanus niloticus* and *Tiliqua scincoides* (*Rieppel, 1979*; *Estes & Williams, 1984*), and high-fibered fruit, such as *Gallotia galloti* (*Valido, Nogales & Medina, 2003*), possess pleurodont dentition, so it cannot be assumed that acrodont ankylosis evolved as a means to process tough foods (or that it is the only means by which to process tough foods, see section below). We also doubt that acrodont ankylosis first arose in response to a particular diet, because basal rhynchocephalians possessing acrodont dentition were likely insectivorous (*Evans, 1980*; *Fraser & Walkden, 1983*; *Whiteside, 1986*). Insectivory is also widespread among extant squamates, which mostly possess pleurodont dentition, so it seems unlikely that the initial evolution of acrodont ankylosis would be strongly influenced by an insectivorous diet. Acrodonty remained widespread as rhynchocephalians diversified to fill various dietary niches (*Jones,*

*2006*; *Jones, 2009*), so it seems unlikely that acrodont ankylosis evolved in response to any particular diet. Possessing firmly ankylosed acrodont dentition in conjunction with a higher bite force does allow access to harder foodstuffs, but it cannot be assumed to be the sole reason for the evolution of acrodont ankylosis.

The second hypothesis is that shared oral mechanics shaped the evolution of acrodont ankylosis, implying a mechanical constraint influenced the evolution of the trait. Previous bite force measurements of *S. punctatus* only measure the orthal bite, and not the force of the propalinal stroke (*Schaerlaeken et al., 2008*; *Jones & Lappin, 2009*). While both squamates and *Sphenodon punctatus* are capable of orthal shearing, *S. punctatus* is well-known for possessing an akinetic skull and using propalinal jaw movement, where the lower jaw moves in an anterior-posterior motion (*Robinson, 1976*; *Gorniak, Rosenberg & Gans, 1982*; *Cartland-Shaw et al., 1998*; *Jones, 2008*). This is in contrast with most squamates which possess kinetic skulls and typically favor streptostyly in order to move the lower jaw in a fore and aft motion (*Evans, 2008*). The rigidity afforded by an akinetic skull does allow for a relatively stronger orthal bite in *Sphenodon punctatus* than most lizards, and kinesis is thought to reduce the strength of a bite though allowing for improved prey capture and handling. However, the acrodont taxa examined here possess some of the least kinetic skulls among squamates (*Iordansky, 1990*; *Arnold, 1998*), possibly improving the capability of a strong bite. Though the acrodont taxa examined here do possess more rigid skulls, allowing for stronger biting, there does not seem to be a shared oral mechanism that would influence the evolution of acrodont ankylosis.

It is worth noting that not all acrodontan lizards possess a fully acrodont dentition. Agamid lizards possesses a dentition that is anteriorly pleurodont and posteriorly acrodont (*Cooper, Poole & Lawson, 1970*). While the pleurodont dentition still regularly replaces, the acrodont dentition does not. In those taxa, tooth morphology varies greatly along the tooth row, with the anterior pleurodont teeth being more slender and sharp. Bite force decreases as gape angle increases (e.g., *Dumont & Herrel, 2003*), thus the anterior pleurodont dentition is consistently subjected to less force than the posterior acrodont dentition and need not be as 'reinforced' as the posterior acrodont dentition. This combination of factors (tooth morphology, implantation, attachment, and relative position) suggests that the anterior and posterior portions of the mouth play differing roles in food processing. While the anterior dentition may be better suited for grasping and piercing, the posterior dentition is more suited for crushing. Chameleons do not possess such a marked difference in tooth morphology, implantation, and attachment as agamid lizards. This could reflect a difference in diet as well as a difference in the nature of lingual prey apprehension between agamids and chameleons. Though both use lingual prey apprehension to capture food, chameleons can ballistically project their tongues a considerable distance in comparison to agamids (*Meyers & Nishikawa, 2000*). This projection could certainly affect, or potentially damage, the anterior dentition if it were of a more slender morphology and not strongly ankylosed to the jaw bone.

Of the taxa that were examined in previous publications, fewer species of lepidosaurs with acrodont tooth implantation have been studied in regard to bite force in comparison to those with pleurodont implantation (Fig. 5). Of the pleurodont taxa, 49 species of

*Anolis* lizards (Dactyloidae) were the subjects of 20 publications that record bite force alone. Those taxa make up the largest proportion of pleurodont taxa analyzed here. The large number of *Anolis*-based studies is partly because they are speciose and represent a particularly important model taxon for ecological and evolutionary studies in the Americas. Conversely, only 16 unique species of acrodont lepidosaurs belonging to four separate families are the subject of 17 publications that record bite force. Only 16 families of squamate lizards have been subjected to bite force analyses, which leaves a large portion of the squamate line understudied (Fig. 5). Further examination of bite force and diet across Lepidosauria may enforce our hypothesis while also revealing other ecological and evolutionary trends.

## Can acrodont ankylosis be reversed?

The transition from pleurodont to acrodont tooth implantation occurred independently several times within Lepidosauria (Acrodonta, Trogonophidae, Rhynchocephalia) and it is even seen in stem lepidosauromorphs (*Sobral, Simões & Schoch, 2020*), but only in Acrodonta and Rhynchocephalia is the tooth-bone boundary difficult to detect upon initial inspection. Stem Acrodonta do not possess the extensive bone deposition that accompanies ankylosis, nor do they possess the apical tooth implantation that is seen in the crown group, although the roots of the teeth are much shorter than most other iguanian lizards and possess a relatively increased degree of ankylosis (*Simões et al., 2015*). All crown acrodontans possess some degree of acrodont tooth implantation accompanied by ankylosis and bone deposition. Within that clade, acrodont ankylosis may appear to be a fixed trait that lacks the plasticity to explore other forms of tooth implantation and attachment. However, developmental evidence suggests that the tissues that promote tooth formation and replacement in some acrodontan lizards is still present (*Salomies et al., 2019*), thus there is potential for the reversal of monophyodonty, and by extension, strong ankylosis. The more likely explanation is that there has been no selective pressure acted upon tooth implantation, attachment, and replacement within Acrodonta that would drive members of the clade away from acrodont ankylosis since it initially evolved. While this may imply a potentially adaptive circumstance to the initial evolution of this trait, it cannot be excluded that this trait may no longer act as an adaptation in extant Acrodonta.

Acrodont ankylosis is persistent within Rhynchocephalia, but several transitions in tooth implantation occurred from an initially acrodont state (*Jenkins et al., 2017*). *Ankylosphenodon pachyostosus* possesses 'ankylothecodont' dentition, in which the tooth has deeply implanted roots, but is nonetheless ankylosed to the surrounding bone (*Reynoso, 2000*). One genus, *Sapheosaurus*, potentially lacks marginal dentition, although it is unknown if this is due to extensive wear or if this taxon was truly edentulous (*Cocude-Michel, 1963*). The teeth of *Priosphenodon avelasi* possesses a more complex dentition, with teeth set within a shallow socket and periodontium holding adjacent teeth together (*LeBlanc et al., 2020*). The tooth plates seen in *Oenosaurus muelheimensis* also represent an interesting derivation from the typical tooth seen within Rhynchocephalia (*Rauhut et al., 2012*). Nonetheless, the tooth implantation of *O. muelheimensis* was described as acrodont. Although acrodonty is widespread within Rhynchocephalia, tooth implantation

seems to be a more plastic trait within this clade than it is within Acrodonta. Though implantation and attachment within Rhynchocephalia may vary, monophyodonty seems to be persistent in derived rhynchocephalians. However, the most basal rhynchocephalian known, *Gephyrosaurus bridensis*, did replace its pleurodont teeth regularly much like pleurodont lizards (*Evans, 1985*).

## More than one way to crush a clam—durophagous pleurodonts

Aside from the state of acrodont ankylosis, other forms of dentition may act in a similar function. Many suggest that the molariform teeth of durophagous lizards are well equipped for withstanding strong, crushing bites necessary for ingesting molluscs and other hard-shelled prey (*Evans & Sanson, 1998*; *Schaerlaeken et al., 2012*). *Dracaena guianensis* and *Tiliqua scincoides* (both pleurodont) are the only durophagous taxa for which SVL-NBF data could be analyzed within the present study. *D. guianensis* is the only durophagous representative for HD-NBF. Though sample size is limited, those durophagous taxa showed the lowest median NBF values (Figs. 3 and 4). However, raw data reports a bite force of 383.3 N for *D. guianensis*, which is among the higher raw bite force values recorded. Given the general increase in bite force with size, the larger overall size of *D. guianensis* compared to most taxa within the dataset (Supplemental Information) is likely the primary driver of its high bite force, although other morphological, evolutionary, and ecological factors may play supporting roles. Nonetheless, even under higher bite forces, their teeth are pleurodont, suggesting that not all dentition need evolve into acrodont ankylosis in order to withstand high bite forces.

*Varanus niloticus* has been subjected to several studies concerning its dental morphology and cranial kinematics (*Rieppel, 1979*; *Rieppel & Labhardt, 1979*; *Condon, 1987*; *D'Amore, 2015*). *V. niloticus* undergoes an ontogenetic change in dentition, with juveniles possessing more slender teeth that later transition to more bulbous molariform dentition (*Rieppel & Labhardt, 1979*; *D'Amore, 2015*). This ontogenetic shift in tooth morphology is often attributed to an ontogenetic shift in diet, with adults consuming larger proportions of molluscs and crabs (*Rieppel & Labhardt, 1979*; *Luiselli, Akani & Capizzi, 1999*; *Lenz, 2004*). However, some suggest there is no evidence for a dietary shift within this species (*Bennett, 2002*), while others show that certain populations consume snails and crabs while other populations do not consume hard-bodied prey (*Losos & Greene, 1998*). Other species of *Varanus* without specialized dentition are also known to eat hard-bodied prey, such as turtles and crabs (*Losos & Greene, 1998*). We might presume *V. niloticus* has a relatively high raw bite force, allowing for the consumption of hard-bodied prey, but how that relates to body size and how it compares to the bite forces of other varanids is unknown.

## Acrodonty in Amphisbaenia

We know little about tooth attachment in Trogonophidae, though the clade is thought to be acrodont and the teeth are likely ankylosed (*Gans, 1960*; *Gans & Montero, 2008*). The relatively lower bite force seen in *T. wiegmanni* compared to other acrodont taxa seen in our results was likely impacted by the fact that *T. wiegmanni* is an elongate, serpentine-like form. Because of that, using SVL to standardize our results may not be meaningful in the

case of this taxon. However, the other taxa examined in this study are not impacted by extremely elongate body plans. When bite-force is standardized by head depth, the same trend of greater acrodont bite force is more apparent for *T. wiegmanni*. Further work on the dentition of this clade would clarify if it too possesses strong ankylosis or bone deposition around the base of the dentition like that of Acrodonta and *Sphenodon punctatus*.

Trogonophidae is the only clade within Amphisbaenia to evolve acrodont tooth implantation. However, other amphisbaenians possess teeth with roots of varying lengths. Overall, amphisbaenians possess shorter roots than what is seen in most iguanians and geckos. Tooth implantation in amphisbaenians is often described as 'subacrodont' or 'subpleurodont' to denote the stray from the 'typical' pleurodont tooth implantation seen in most other squamates (*Estes, 1975*; *Yatkola, 1976*; *Sullivan, 1985*; *Charig & Gans, 1990*; *Kearney, Maisano & Rowe, 2004*; *Gans & Montero, 2008*; *Longrich et al., 2015*; *Čerňanský, Klembara & Müller, 2016*). The evolution of tooth implantation and attachment in Amphisbaenia has not been explored further, but the trend towards dentition with shorter roots is intriguing. Bite-force experiments conducted on amphisbaenians could address if the evolution of acrodonty within the clade is related to high bite force and diet. However, we cannot exclude the evolution of acrodont tooth implantation within Amphisbaenia may have arose for other reasons, such as limited jaw space.

## CONCLUSIONS

Acrodont ankylosis accompanied by increased bone deposition seen in Acrodonta and *Sphenodon punctatus* is likely an adaptation related to strong bite force. We do not know if this form of tooth implantation and attachment evolved in response to high bite force, or *vice versa*. Nonetheless, there are behavioral implications for the early evolution of this trait. Changes in tooth implantation and attachment are often associated with diet-related hypotheses. However, it cannot be presumed that increased bite force or changes in tooth implantation and attachment are only associated with diet. When discussing the evolution of such traits, we must take into account other possible behavioral influences, such as territory defense, intraspecific combat, and mating success, which are also associated with increased bite force. Testing such hypotheses in the fossil record may prove impossible, but it is still necessary to speculate all scenarios. Furthermore, these dental traits may have evolved convergently in response to different selective pressures depending on the clade. Acrodont ankylosis accompanied by bone deposition may be fixed traits in Acrodonta, which has not explored other forms of tooth implantation and attachment. However, rhynchocephalians were able to explore other forms of tooth implantation throughout their evolutionary history, though are largely monophyodont.

Acrodont ankylosis is not the only form of tooth implantation and attachment potentially associated with higher bite forces. Durophagous squamates, though pleurodont, often possess molariform tooth morphologies that are also able to withstand increased bite force. In the present study, it appears that higher bite force is likely related to larger body size for these specialists. However, bite force and its relationships with tooth morphology requires further study in durophagous squamates as well as other dietary specialists. Most of the

taxa analyzed were insectivorous generalists, though carnivorous acrodonts, herbivorous acrodonts and pleurodonts, and durophagous, frugivorous, and omnivorous pleurodonts were also included. The array of diets seen in extant acrodont taxa suggests that *if* acrodont ankylosis evolved as an adaptation to a particular diet, it may no longer act in an adaptive capacity for a specific diet. We encourage further study on dietary specialists, for greater variation in bite force may exist among Squamata, with subsequent implications for dental evolution in terms of tooth implantation, attachment, replacement, and morphology.

## ACKNOWLEDGEMENTS

We thank D. Meyer, M. Fabbri, and R. Armfield for providing helpful comments on an early version of this manuscript. Access to specimens was aided by G. Watkins-Colwell at the Yale Peabody Natural History Collections. We thank the three reviewers of this manuscript, Y. Haridy, A.R.H. LeBlanc, and K.S. Brink, who provided helpful comments and additional suggestions that greatly enhanced this work.

### Funding
This work was supported by the Yale Peabody Museum John T. Doneghy Fund. The funders had no role in study design, data collection and analysis, decision to publish, or preparation of the manuscript.

### Grant Disclosures
The following grant information was disclosed by the authors:
Yale Peabody Museum John T. Doneghy Fund.

### Competing Interests
The authors declare there are no competing interests.

### Author Contributions
- Kelsey M. Jenkins conceived and designed the experiments, performed the experiments, prepared figures and/or tables, authored or reviewed drafts of the paper, and approved the final draft.
- Jack O. Shaw conceived and designed the experiments, analyzed the data, prepared figures and/or tables, authored or reviewed drafts of the paper, and approved the final draft.

### Data Availability
The raw data, the R script, and all studies from which the data were collected for this meta-analysis are available in the Supplementary Files.

### Supplemental Information
Supplemental information for this article can be found online at http://dx.doi.org/10.7717/peerj.9468#supplemental-information.

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
