# Peer review of "Bite force data suggests relationship between acrodont tooth implantation and strong bite force"

_PeerJ, doi:10.7717/peerj.9468_

## Round 0.1 · original submission · Major Revisions

Your work appears to be preliminary in nature and therefore limited in its conclusions. A more thorough examination of the issue is required.
Please, together with your unmarked revised manuscript, provide a marked-up copy as well as a document explaining how you have addressed each of the points raised by the reviewers.

·

Basic reporting

no comment

Experimental design

no comment

Validity of the findings

no comment

Additional comments

I enjoyed reading this paper which proposes an interesting hypothesis backed by convincing data, and absolutely is deserving of publication after what I perceive to be minor but important alterations. I do not wish to remain anonymous, I am Yara Haridy the authors are free to contact me with further questions ([email protected])

First, the things I thoroughly appreciated:
• Clear concise writing and getting to the point
• We need more ‘why did so and so evolve?’ papers that actually go out and test the idea!
• I really enjoyed the section about “can ‘AVSA’ be reversed?” I’ve considered exactly this, and I too think that Acrodontians basically pigeonholed themselves into this adaptation early on and were unable to adapt or diversify further. I agree with the authors that perhaps this is due to an adaption by their ancestor that is no longer as starkly present in current surviving taxa, great points.

Issues from most to least importance:
• The acronym of AVSA is not my favorite, however if the authors wish to keep it, I encourage further explanation. I sympathise with the need to easily refer to this specific type of acrodonty however saying ‘Severe ankylosis accompanies acrodonty’ and then saying ‘acrodonty via severe ankylosis’ sounds contradictory and the latter makes it sound like the severe ankylosis causes the acrodonty. I partially agree that the mode of attachment actually changes the ‘look’ of implantation (Haridy, 2018)
• It is worth mentioning why you chose the term ‘sever ankylosis’ and actually explain how much remodeling and bone invasion there is of the tooth space and why this is not normal ankylosis (Haridy, 2018). This actually lends itself to your argument.
• One issue is the delineation of implantation vs mode of attachment. Many animals can be considered acrodont or sub-acrodont in implantation (i.e. snakes) but are not traditionally thought of having the same specialized convergent dentition as Acrodontians and Rhynchocephalians. The two aforementioned groups have an amazing convergent trifecta in common, position of teeth, lack of replacement, and extreme ankylosis. I feel like this can be emphasized throughout the manuscript. The authors do a fine job of explaining the modes of implantation but not that of attachment, i.e gomphosis vs ankylosis. Teeth fused to the jawbone are likely ‘sturdier’ than those attached via soft tissue. It should also be noted that most animals which have ankylosed teeth go from a gomphosis to an ankylosis at varying rates (see LeBlanc et al. 2016).
• Overall my biggest criticism is that I want ‘more’ I found this paper lacking in depth in many of the exceptional points made by the authors. This is not a major issue but adding more to certain points like (line 163-164) “acrodont tooth with a wide base and firm attachment is better suited to withstand higher bite forces” this should go into how and why the fused attachment is better than ligamentous attachment. In the PDF I outline where I’d like areas expanded on and I think it would greatly help in convincing your reader of your hypothesis.
• I also feel like this paper did not propose alternate hypothesis for the evolution of acrodonty. the proposal of alternate hypothesis then shutting them down systematically would have greatly aided the authors’ biteforce point. An example of an alternate hypothesis could include ‘dietary adaptation’ which is easily refuted by the fact that acrodontians-at least modern ones are varied in their diets as much as non acrodontians so it is unlikely for acrodonty to have evolved for such a reason.
• While bite-force data is one way to show that ‘acrodonty with ankylosis’ evolved to withstand (or because of) increased force another possibility for testing this idea is via FEA to actually see if the acrodont tooth shape/attachment and uniformity of material interface actually increases the force that can be taken by the tooth structures. Of course I am not suggesting that the authors undergo this lengthy study I am only suggesting that perhaps in can be part of a future directions if the authors wish to include it.

·

Basic reporting

The basic reporting is clear, given the available data from the literature. I would suggest that the authors call this study a Meta-Analysis somewhere in the title and Materials and Methods, though. There are no experiments or direct observations in this study.

I would add that for clarity, the authors should also add a figure of what they think constitutes an acrodont tooth without AVSA. At this point, I do not see why we need to introduce the term AVSA for acrodont reptiles (see the attached PDF for more specific comments on this).

Experimental design

One experimental design issue that I can see is that the data aren't segregated by diet. How is the data set skewed in terms of herbivorous/omnivorous/carnivorous/durophagous lizards for either group? Are the acrodont and pleurodont study groups comparable in the breadth of diets? This should be explored further. Also, show the reader where some of the taxa of interest plot in your figures.

I would also recommend that someone who is more adept than I go through your statistical analyses.

Validity of the findings

My major comment would be that this is a very preliminary correlative study. As a result, the conclusions are quite superficial. Given the proposed correlation between higher bite force and acrodonty, I would like to see far more discussion of why all of the durophagous and more robust-toothed lizards probably do not come from acrodont ancestors. This is particularly true for Varanus niloticus, where I don’t see any evidence of a major shift in tooth implantation between it and any of its relatives. They all look pretty pleurodont to me. So why are these hard-biting or shell-crushing lizards not acrodont? Pleurodonty seems to work just fine for these diets.

I am also unclear as to how AVSA affects the integrity of a tooth. The term seems redundant to me, given that the wide base of the tooth and its position on top of the jaw are principle factors the authors cite as potential adaptations to (or consequences of) high bite forces. These are features of tooth implantation (acrodonty), not degree of attachment (ankylosis). To me, AVSA is synonymous with "acrodont ankylosis" here. Teeth fused to apex of jaw.

Additional comments

Please see my attached PDF for all of the specific comments and suggested edits.

---

## Round 0.2 · Minor Revisions

Reviewers 2 and 3 have made additional suggestions and comments. Please, take them into account during the revision of your manuscript.

·

Basic reporting

Clear, fairly unambiguous, professional writing. I have made some suggestions for clearer reporting in the Materials and Methods. These are minor changes.

Experimental design

Again, clear, easy to follow. I like the scrutiny of the data in the discussion. Much more thorough treatment of the data on the whole this time.

Validity of the findings

Funny thing: I feel as though this revised version carries more weight, not just because of the "positive" results, but the falsified hypotheses as well. It now serves as a reference for just how complex the relationships between tooth characters, body size, and diet are. I have no major issues here.

Please see the attached PDF for all of my minor comments.

Additional comments

Much improved in my opinion. Thanks for trying to appeal to the "Meta-Analysis" concept. This is a very common type of study in (e.g.) ecology (not sure where PeerJ is getting the idea that it is strictly a biomedical article type), but that is a moot point at this stage. Everything is fairly clear in the manuscript.

All of my comments are minor, but I hope you take a look through the attached PDF and find them useful in clarifying a few key sections.

·

Basic reporting

This paper is clear and well-written. There are a few references that could be added, see below.

Experimental design

Methods are appropriate for this study.

Validity of the findings

All data are provided, conclusions are well-stated.

Additional comments

This paper is a well-written discussion of the possible functional origins of acrodonty. All previous reviewer comments have been addressed. I had a few ideas and comments while reading the manuscript, listed by PDF line numbers.

Line 40: add reference to Salomies et al., 2019 for development of acrodont dentitions.

Line 45: I think it is known that anchoring the teeth does not inhibit tooth replacement. This is discussed in Haridy (2018). The teeth become anchored after tooth replacement stops. This is also evident in the developmental evidence, where the tissues that make teeth stop making teeth, then the bone develops (Buchtova et al., 2013, Dosedelova et al., 2016, Handrigan and Richman 2010, Salomies et al., 2019).

Line 85: Preuschoft 1991 is a good reference here, since they actually test the function of plicidentine. Also see discussion in Brink et al., 2014.

Line 204: The evolution of the periodontal ligament is also associated with a decrease in tooth replacement rates in the mammalian lineage (see discussion in LeBlanc et al., 2018).

Lines 236-255: In animals that have acrodont teeth at the back of the jaws and pleurodont teeth at the front, the teeth are usually different shapes. Bite force can change depending on where it is measured around the mouth (related to gape), and bite force or bite efficiency changes with tooth shape (e.g., Dumont and Herrel 2003, Huber et al. 2009, and I’m sure there are many more). It should be discussed how in some species, like Agama and Pogona, there is a reason why polyphyodonty is maintained in the anterior teeth, which could be related to bite force, tooth function, heterodonty, jaw function, etc. (see discussion on tooth function in Agama in Cooper et al, 1970). Why does a bearded dragon need pointy anterior teeth while a chameleon does not?

Lines 295-300: From a developmental perspective, there are different ways that acrodont reptiles maintain their dentition. The successional lamina (the tissue that makes the replacement teeth) diminishes after eruption of the first teeth. However, it was recently shown in the bearded dragon that tissue is maintained even after the acrodont teeth have erupted, even though no new teeth are made in the acrodont areas (Salomies et al., 2019). This is interesting when considering your discussion on whether or not acrodonty is reversible. It is possible that teeth could be replaced in the acrodont areas with a few genetic tweaks of that tooth-making tissue. The potential for reversing monophyodonty is there. However, there would need to be a functional reason for this reversal to happen in the evolution of an acrodont lineage.

Line 356: there are pleurodont crushing teeth in Amphisbaena (Pregill 1984).


References:
Brink KS, LeBlanc ARH, Reisz RR. 2014. First record of plicidentine in Synapsida and patterns of tooth root shape change in Early Permian sphenacodontians. Naturwissenschaften. 101:883–892.

Buchtová M, Zahradníček O, Balková S, Tucker AS. 2013. Odontogenesis in the veiled chameleon Chamaeleo calyptratus. Archives of Oral Biology. 58(2):118-133.

Cooper J, Poole D, Lawson R. 1970. The dentition of agamid lizards with special reference to tooth replacement. Journal of Zoology. 162(1):85-98.

Dosedělová H, Štěpánková K, Zikmund T, Lesot H, Kaiser J, Novotný K, Štembírek J, Knotek Z, Zahradníček O, Buchtová M. 2016. Age‐related changes in the tooth–bone interface area of acrodont dentition in the chameleon. Journal of Anatomy. 229(3):356-368.

Dumont ER, Herrel A. 2003. The effects of gape angle and bite point on bite force in bats. Journal of Experimental Biology. 206(13):2117-2123.

Handrigan GR, Richman JM. 2010. Autocrine and paracrine shh signaling are necessary for tooth morphogenesis, but not tooth replacement in snakes and lizards (Squamata). Developmental Biology. 337(1):171-186.

LeBlanc ARH, Brink KS, Whitney MR, Abdala F, Reisz RR. 2018. Dental ontogeny in extinct synapsids reveals a complex evolutionary history of the mammalian tooth attachment system. Proceedings of the Royal Society B. 285(20181792):1-9.

Pregill G. 1984. Durophagous feeding adaptations in an amphisbaenid. Journal of Herpetology.186-191.

Preuschoft H, Reif W-E, Loitsch C, Tepe E. 1991. The function of labyrinthodont teeth: Big teeth in small jaws. In: Schmidt-Kittler N, Vogel K, editors. Constructional morphology and evolution. Berlin: Springer. p. 151-171.

Salomies L, Eymann J, Khan I, Di-Poï N. 2019. The alternative regenerative strategy of
bearded dragon unveils the key processes underlying vertebrate tooth renewal. Elife. eLife 2019;8:e47702.

---

## Round 0.3 · accepted · Accept

Manuscript accepted for publication.

·

Basic reporting

no comment

Experimental design

no comment

Validity of the findings

no comment

Additional comments

This is a great paper discussing the functional reasons for the evolution of acrodont dentitions. It is multi-faceted, considering many forms of evidence to support the hypothesis that acrodont ankylosis is an adaptation allowing for (or the result of) a stronger bite force. All previous reviewer comments have been addressed appropriately, and I have no further comments.